# Focus group discussions on low-flow oxygen and bubble CPAP treatments among mothers of young children in Malawi: a CPAP IMPACT substudy

Kristen L Sessions [ORCID],[1] Laura Ruegsegger,[2] Tisungane Mvalo,[3,4] Davie Kondowe,[3] Mercy Tsidya,[3] Mina C Hosseinipour,[3,5] Norman Lufesi,[6] Michelle Eckerle,[7,8] Andrew Gerald Smith,[9] Eric D McCollum[10,11]

For numbered affiliations see end of article.

**Correspondence to**
Dr Kristen L Sessions;
ksessions@luriechildrens.org

## ABSTRACT

**Objective** To determine the acceptability of bubble continuous positive airway pressure (bCPAP) and low-flow oxygen among mothers of children who had received either therapy.

**Setting** A district hospital in Salima, Malawi.

**Participants** We conducted eight focus group discussions (FGDs) with a total of 54 participants. Eligible participants were mothers of children 1 to 59 months of age with severe pneumonia and a comorbidity (HIV-infection, HIV-exposure, malnutrition or hypoxaemia) who, with informed consent, had been enrolled in a randomised clinical trial, CPAP IMPACT (ImprovingMortality for Pneumonia in African Children Trial), comparing low-flow oxygen and bCPAP treatments (ClinicalTrials.gov, NCT02484183).

**Primary and secondary outcome measures** FGDs assessed mothers' attitudes and feelings towards oxygen and bCPAP before and after therapy along with general community perceptions of respiratory therapies. Data was analysed using inductive thematic analysis to assess themes and subthemes of the transcripts.

**Results** Community perceptions of oxygen and bCPAP were widely negative. Mothers recounted that they are told that 'oxygen kills babies'. They are often fearful of allowing their child to receive oxygen therapy and will delay treatment or seek alternative therapies. Mothers report limiting oxygen and bCPAP by intermittently removing the nasal cannulas or mask. After oxygen or bCPAP treatment, regardless of patient outcome, mothers were supportive of the treatment their child received and would recommend it to other mothers.

**Conclusion** There are significant community misconceptions around oxygen and bCPAP causing mothers to be fearful of either treatment. In order for low-flow oxygen treatment and bCPAP implementation to be effective, widespread community education is necessary.

## Strengths and limitations of this study

► Focus group discussions included mothers of surviving or deceased children who had severe pneumonia and received either oxygen or bubble continuous positive airway pressure (bCPAP) therapy.

► This study was part of a larger randomised control trial, CPAP IMPACT (Improving Mortality for Pneumonia in African Children Trial), which sought to compare hospital outcomes between bCPAP and low-flow oxygen recipients with severe clinical pneumonia.

► Groups were facilitated by a trained Malawian facilitator and data was collected until thematic saturation was reached, suggesting sufficient sample size for addressing acceptability.

► This study is limited in that it may not be representative of other community beliefs regarding healthcare and mothers who did not present to the hospital or declined focus group participation were not included. Additionally, all focus groups were conducted post-intervention.

with low blood oxygen saturation—hypoxaemia—is low-flow oxygen. Malawi initiated the Child Lung Health Programme in October 2000, which installed oxygen concentrators in each district hospital and trained healthcare workers in proper indications for oxygen use.[2 3] Despite these efforts, mortality remains high and there may be a need for advanced treatment modalities to further decrease child mortality from pneumonia.

Bubble continuous positive airway pressure (bCPAP), a form of non-invasive ventilation that provides continuous positive end expiratory pressure to spontaneously breathing patients, has been shown to reduce mortality in neonates with respiratory distress.[4 5] However, evidence of effectiveness of bCPAP among older children is less conclusive.[6] Although the effectiveness of bCPAP in this

## INTRODUCTION

Pneumonia remains a leading cause of death in young children worldwide with 50% of mortality occurring in sub-Saharan Africa.[1] Current standard of care for children in low-income African countries, such as Malawi,

BMJ

age group is uncertain, implementation is ongoing in many low-income and middle-income countries including Malawi, Kenya, Ghana, Honduras and Malaysia.[6–10]

For an existing technology like oxygen and a new technology such as bCPAP to be most effective, they must be feasible to administer by healthcare personnel and acceptable to caregivers so that treatment adherence is optimised. Previous studies have demonstrated that bCPAP use on infants can be stressful and frightening to mothers.[11–13] In-depth interviews found mothers of neonates receiving bCPAP in Malawi were anxious and fearful, most indicating that they received inadequate information about the treatment.[11] Commonly quoted sources of stress include tubing, decreased ability for physical contact and limited parent-child interactions.[12 14] In this study, we sought to determine if mothers of children 1 to 59 months of age with pneumonia found oxygen and bCPAP acceptable. We also sought to identify community knowledge and feelings toward various treatment options for pneumonia.

## METHODS
### Study design
In February and March of 2018, we conducted eight focus group discussions (FGDs). Groups consisted of 6 to 10 mothers of children 1 to 59 months old who had previously received care for pneumonia as part of the randomised controlled trial CPAP IMPACT (Improving Mortality for Pneumonia in African Children Trial; ClinicalTrials.gov, NCT02484183), which sought to compare hospital outcomes between bCPAP and low-flow oxygen recipients with severe clinical pneumonia who, based on WHO recommendations, would benefit from supplemental oxygen therapy.[15] Prior to enrolment trial participants underwent a process of informed consent, which reviewed and demonstrated bCPAP and oxygen care, bCPAP devices including masks, prongs and tubing, low-flow oxygen nasal cannulas, study blood draws and also expected routine hospital care related to both modalities not limited to nasogastric tube feeding, intravenous fluids, blood draws and nasal suctioning. The benefits and risks of trial participation were also reviewed during consent. Participants were permitted to withdraw from the main trial at any point. Focus groups were stratified by the trial's oxygen or bCPAP study groups and also whether the child survived or died in the hospital.

FGDs were held in a private room at Salima District Hospital (SDH) and were conducted in Chichewa, the local language, by a trained facilitator. Information about perceptions, education and reactions toward bCPAP or oxygen were elicited. Mothers were asked to discuss beliefs they had heard from their communities, information learnt through the study consent process and experiences during their child's treatment. Each session lasted for about 1 hour and was audio-recorded along with detailed notes taken by an independent note taker.

A debriefing session with the study team took place after each session.

### Sampling
Mothers were chosen by opportunistic and convenience sampling by contacting mothers whose child had been enrolled in CPAP IMPACT beginning with the most recent discharges. All mothers whose child had been enrolled in CPAP IMPACT were eligible to participate in a FGD. Women who were aged 16 years old and above but had a child were allowed to participate in accordance with Malawi ethical research standards and laws. They were contacted via phone and invited to return to the hospital to participate. Contact information was missing for more mothers whose child died than whose child survived. However, among mothers who were successfully reached, no-show rates for FGDs were similar between mothers whose child survived and those who died. Mothers received a travel stipend and snack for participation in the FGD, which is research participation compensation in accordance with Malawi ethical standards.

### Data management and analytics
Focus group discussions were audio-recorded, transcribed, translated from Chichewa to English and then back translated to ensure meaning was preserved. Chichewa is a gender-neutral language so 'they' or 'them' were used throughout translations. Data was managed using NVivo 11 QSR software. Transcripts were read in full to determine tone and scope of information. They were each coded by two independent reviewers (KS and LR) for themes and subthemes using inductive thematic analysis and a framework analysis approach.[16–18] Initial codes were developed a priori based on the interview guide but an iterative coding process was implemented to allow for flexibility and responsiveness to discussion results (online supplementary appendix 1). Themes, subthemes and connections between concepts were noted and discussed with the team. Statistical analyses for quantitative variables used mean and SD to describe normally distributed data and median and IQR to describe not normally distributed data. We compared means of normally distributed data using t-tests and medians of not normally distributed data using the Wilcoxon-Mann-Whitney test. Quantitative analyses were conducted using Stata 15.1 (College Station, USA).

Written consent was obtained in Chichewa from all participants. Consents were read aloud to participants who were unable to read. Data was de-identified prior to analysis.

### Patient and public involvement
Due to resource limitations, it was not possible to involve patients or the public in the trial design, or conduct, or reporting or dissemination of our research. However, members of the Salima District Hospital administration and Malawi Ministry of Health were involved in the design, conduct, reporting and dissemination of this study.

**Table 1** Demographics of mothers who participated in a focus group discussion

|  | Total (n=54) | bCPAP (n=25) | Oxygen (n=29) | P value |
|---|---|---|---|---|
| Age of mothers in years, median (IQR) | 26 (21 to 32) | 26 (21 to 31) | 25 (22 to 32) | 0.8755 |
| Number of children, mean (SD) | 2.7 (1.5) | 2.7 (1.4) | 2.6 (1.7) | 0.8887 |
| Age of participant children in months, median (IQR) | 7.1 (3.1 to 14.2) | 6.7 (3.1 to 12.1) | 8 (4.8 to 14.2) | 0.6148 |
| Child died during study, n (%) | 11 (20.4) | 5 (20.0) | 6 (20.7) | 0.8580 |
| Straw roof, n (%)* | 43 (79.6) | 21 (84.0) | 22 (75.9) | 0.9635 |
| Completed primary school or higher education, n (%) | 14 (25.9) | 6 (24.0) | 8 (27.6) | 0.7791 |
| Previous exposure to someone who was given oxygen therapy, n (%) | 7 (13.0) | 4 (16.0) | 3 (10.3) | 0.7833 |

*Straw roof compared with a metal sheet roof was used as a marker of socioeconomic status.
bCPAP, bubble continuous positive airway pressure.

## RESULTS

### Demographics

A total of 54 participants took part in one of eight focus groups. Table 1 reports the demographics of mothers who participated and indicates that the characteristics of mothers participating in this substudy were balanced between trial arms. Participants were an average of 26.7 years old and had 2.7 children. Most participants (n=43; 79.6%) had a low socioeconomic status as evidenced by living in a home without a metal roof and 74.1% (n=40) did not complete primary school.

### Major themes

Throughout the focus groups, multiple themes arose including widespread negative beliefs about oxygen and bCPAP in the community, anxiety and fear surrounding care before initiation of treatment, and a shift in beliefs to a more positive opinion of treatment regardless of whether their child lived or died (table 2). Additionally, participants were able to identify potential benefits, harms and alternatives to both oxygen and bCPAP.

### Knowledge and feelings about oxygen and bCPAP prior to treatment

Despite longstanding use of oxygen in Salima District Hospital, knowledge of oxygen, and also bCPAP, was low within the participants' communities. Before participation in this study, only 13% (7/54) of participants personally knew someone who had received oxygen therapy. Despite limited personal exposure to oxygen, all mothers had heard of oxygen from members of their communities. Almost all information participants received from the community was negative. Eighty-three per cent of mothers (45/54) explicitly, and without prompting, mentioned hearing negative opinions of supplemental oxygen therapy in the community, many using the phrasing that 'oxygen kills' (table 2).

"At the village, people say that oxygen kills babies. So, when they told me that my baby will be on oxygen, I was very much worried about it." (oxygen, age 19, child survived)

They were also told to refuse therapy. One mother described her experience by saying,

"People back in the village tell you to refuse the oxygen when you come to the hospital because they say a child who is put on an oxygen machine dies easily." (oxygen, age 33, child survived)

Not only are negative things said in the community, but also within the hospital. Mothers often reported being told to remove the equipment temporarily or permanently by other caregivers and this led to several self-reports of interrupting care.

"When you are in that room, people stop what they were doing and start looking at you. Some (mothers) say 'I thought they are supposed to be removing it when the doctor is not there?'… They say because (bCPAP) kills." (bCPAP, age 33, child survived)

Some patients reported observing mothers abscond with their child. One mother also reported having her child taken off therapy by a relative.

"One mother ran away with her sick child… She was afraid that oxygen kills. People were telling her that oxygen kills." (oxygen, age 24, child survived)

The knowledge about bCPAP was significantly lower; most reported they had never heard of bCPAP before being educated by the study staff at the hospital. However, those who had heard of bCPAP reported similar negative concern that it can worsen the child's condition or kill the child.

"People in the community say that when a child is on bCPAP, they do not feed because the nose is covered and the child cannot swallow and they say it doesn't take long before the child dies" (bCPAP, age 35, child died)

In both groups, community beliefs led mothers to feel anxious and fearful prior to treatment initiation. Additional sources of fear included lack of familiarity with the machine and the severity of illness of the other children receiving treatment.

**Table 2** Major themes that emerged

| | | |
|---|---|---|
| Community knowledge | Overall low knowledge, with less knowledge about bCPAP compared with oxygen | "We are coming from the communities where we have no knowledge about this (bCPAP)." |
| | Negative thoughts about supplemental oxygen: 'oxygen kills' | "In our community, we are told that when someone is on oxygen they die so quickly because oxygen increases the death pace." |
| Feelings before treatment | Almost all very anxious due to things said in the community or the appearance of the oxygen cannula or bCPAP machines and face masks | "People say oxygen kills and this makes people feel afraid of the treatment. Someone ran away because she was afraid of the oxygen." |
| Alternative treatments | Traditional medicine includes use of herbs and 'traditional medicine tattoos' | "My granny tried to help my baby using herbs… She used some to rub herbs against their ribs, and some she made them drink, but it never worked out." |
| | Baby should be kept warm and environment clean | "The main cause of pneumonia is cold so if the child is kept warm it helps in that no coldness can penetrate them again." |
| | Injections or paracetamol can be given at health centres | "When a child has suffered from pneumonia and you go to the hospital, the doctors give the child injection." |
| Feelings after treatment | Caregivers remained supportive of both oxygen and bCPAP regardless of child's outcome | "Oxygen helps but where the condition is not reversible, it does not. It takes someone to be lucky but it is helpful." |
| | Frequently stated that they would encourage others to seek treatment | "I am capable of encouraging them… I tell them that if a child has pneumonia, take them to the hospital and if they want to give them bCPAP, accept it." |
| Benefits to treatment | Helps the child breathe, allows child to feed and become more active and aids in the child's survival | "I felt good because my child was able to feed while they had the oxygen tubes in the nose." |
| Harms to treatment | Majority had no concerns, though some worried about bruising, rashes and nasal injury and bCPAP machine malfunctions | "I don't think oxygen therapy can harm a child in any way because the oxygen that is found in the machine is the same that we breathe normally." |
| Education timing | Majority prefer information to be given before treatment | "It is not practical to just put the child on bCPAP before giving information to the mother… people say that this thing kills and if that can happen, the mother can be so afraid and maybe she can run away." |
| | Desire for community sensitisation | "I feel it is better to sensitise the people in the communities prior to their visits to the clinic because the people will have the information already." |

bCPAP, bubble continuous positive airway pressure.

"I was scared because it was my first time to see that thing and I was very worried. When I was entering the room, I found another child dying. So my feeling was that my child was going to die because that is the room where children die." (bCPAP, age 17, child survived)

When one mother was asked about the cause of her anxiety, she responded, "Primarily it was because of what people were saying. But since I had never seen this before, my worry multiplied." (oxygen, age 24, child survived)

Some women did not endorse anxiety around treatment. Instead, women in both groups referred to religious faith and trust in the medical providers as sources of comfort before treatment. These positive opinions were more common among the 13% of women who had had previous firsthand exposure to oxygen therapy.

"I also did not mind what people were saying in the community… I said 'My child will receive this treatment; whatever happens to them; it is the will of God.'" (bCPAP, age 35, child died)

### Alternative treatments for the child's illness

Participants provided several examples of pneumonia treatments that have been used in the community. Many referred to traditional medicine techniques which consist of using herbs to burn and inhale the smoke, to rub on the baby's ribs or to make drinks. Traditional healers may make 'traditional medicine tattoos' in which herbs are placed beneath the child's skin via cuts or 'tie the baby' which refers to making a necklace with a bag of grass and traditional medicine and placing it around the baby's neck or waist.

"When we are at the village, we are told that the best way is to go to the traditional doctors. The traditional doctor makes traditional medicine tattoos and they also provide

other medicines to drink so that the baby should be urinating frequently. After those medicines, they provide other medicines as well that they tie the baby. They say that when the baby is tied, it helps pneumonia to stop." (oxygen group, age 21, child survived)

While multiple complementary medicine techniques were endorsed, the majority of mothers who participated in the FGDs agreed that the best treatment is to seek care at a health centre. Mothers who had tried traditional medicine before commented that they did not see improvement.

"It is just very unfortunate that people go for traditional medicine but the best thing to do is to rush to the hospital." (bCPAP group, age 26, child survived)

Other proposed treatments for pneumonia included keeping the baby warm and medications such as antibiotics and acetaminophen.

### Beliefs and feelings after treatment with oxygen or bCPAP

After treatment, the mothers were supportive of their care regardless of her child's health outcome and perceived multiple benefits of both treatments. Mothers whose child survived on oxygen or bCPAP expressed happiness and gratitude and they would recommend it to others.

"My child was coming here when they were gasping… bCPAP recovered the child. It saved the life of my child and I will never forget. I will proclaim the benefits of bCPAP wherever I will go and I will encourage people to go for this treatment whenever they are offered." (bCPAP, age 35, child survived)

Mothers who had a child die while on therapy were still accepting of treatment. They often referred to other children who improved as evidence for its effectiveness.

"Eventually my child died, not because they were on bCPAP but because it was time for them to die. So I cannot say that bCPAP kills just because my child did not survive; it is helpful because I saw other children surviving but I was just unfortunate that my child died." (bCPAP, age 30, child died)

### Perceived benefits and harms of oxygen and bCPAP

The most commonly discussed benefit was that the machines help children breathe. Feeding was also often used as a marker of health with mothers referencing the ability to feed again as a benefit to both oxygen and bCPAP treatments. Once on treatment, their child became more active. Overall, mothers claimed that the machine helped their child to survive.

"Oxygen is helpful. Even when a child is critically ill but when they put on oxygen the child is able to feed." (oxygen, age 38, child died)

"I feel (bCPAP) increases the oxygen levels and help the child to breathe properly. If the child was not able to feed, they start feeding." (bCPAP, age 30, child died)

A majority of the mothers had no concerns about treatment. Some mothers were concerned about bruising, rashes or nasal injury. Other concerns included machine malfunctions or electricity blackouts, which are common in Malawi.

"My only observation was that after receiving the treatment, my child had red skin meaning that the belt had caused some friction on the skin." (bCPAP, age 35, child survived)

### Education about oxygen and bCPAP

Given the low community knowledge about oxygen and bCPAP, mothers were asked about the best time to be given information about treatment. The majority of mothers preferred information about the treatment to be given before any medical care was provided to the child, as was done in this trial, to address any concerns or allow the mother to decline treatment.

"We are coming from the communities where we have no knowledge about this thing. What people talk about this thing is different from what is on the ground. So it is better to give the information about it before giving treatment so that people are well sensitised on how it works in order to deal with their fears they had from the community." (bCPAP, age 35, child died)

However, several felt it was more important to start the child on treatment first then provide information to the caregiver. This was often the case when they felt the child was very sick and treatment should not be delayed.

"(With) how serious the child is; it is better to start treating the child before giving the information so that the child should get better." (bCPAP, age 26, child survived)

"If you have gone to the hospital sometimes the severity of the illnesses differs …I feel that the proper thing would be to start treating the child first with the oxygen therapy then they can start giving you the information." (oxygen group, age 24, child survived)

Another common theme was the desire for education in the communities. Mothers recognised a need for community sensitisation to feel more comfortable about coming to the hospital.

"I would love if this information is given to us before the child gets sick so that when we come here, it should not be a strange thing to us. They can use the health surveillance assistants to sensitise the communities so that by the time the child will be sick, when we come here, we should not waste time getting the information but instead the child should start treatment right away." (bCPAP, age 26, child survived)

### DISCUSSION

We found that many communities have negative beliefs and fears of supplement oxygen. In the communities, many women have been told that 'oxygen kills' and offered complementary traditional medicine. They have been told to remove the oxygen therapy, which causes interruptions in treatment. Most caregivers were experiencing supplemental oxygen for the first time, and they reported feeling anxious mostly due to the community input.

Regardless of the randomisation group and outcome, after receiving treatment the caregivers reported satisfaction with the treatment and would even encourage others to go to the hospital for care. Mothers of children who died referenced temporary improvement in their child's clinical status or seeing other children improve with treatment as evidence for its effectiveness. This suggests that education and exposure to the treatments, even in the setting of a poor outcome, can be effective.

Many participants saw the benefits of oxygen and bCPAP care, and little saw harm aside from bruising or other marks from wearing the mask. Most mothers prefer information about treatment to be given before it is started and believe that the community would benefit from sensitisation.

Oxygen therapy has been standard of care at SDH for children with severe pneumonia since Malawi implemented oxygen throughout the country beginning in 2000.[3] Despite availability for 18 years, mothers still reported widespread negative beliefs in their communities. However, mothers who had experienced oxygen first-hand reported entering the study with positive feelings. bCPAP had only been available at SDH through CPAP IMPACT for 2 years prior to these focus group sessions. Even though it was less familiar to mothers, those who had heard of it expressed similar fears and beliefs. Mothers of neonates receiving bCPAP in a neonatal intensive care unit in Malawi expressed similar fears and anxieties.[11] Both studies demonstrated satisfaction with the treatment after initiation suggesting that exposure and education can reduce anxiety in mothers of children requiring respiratory support.

The perception that oxygen can kill children could lead mothers to delay seeking care and to seek other treatment options in their communities. Caregivers reported seeing traditional healers or trying home treatments before coming to the hospital. It is unclear how long caregivers delayed seeking more formal healthcare while under the care of traditional healers. They also reported feeling desperate when they arrived at the hospital and willing to accept any treatment given the condition of their child. If only the sickest children receive oxygen, mortality will likely remain high, which may continue to propagate negative beliefs among the community. Even once arriving at the hospital, we found mothers reported continuing to hear negative opinions from other caregivers and were deterred from accepting or continuing treatment. The intermittent removal of the mask or nasal prongs by caregivers during oxygen and bCPAP treatments could interfere with treatment effectiveness and lead to poorer than expected outcomes. Caregivers implied that removal of oxygen and bCPAP treatments occurred when healthcare providers were not present, perhaps while caring for other children or at night when the staff-to-patient ratio was lower. While the frequency of removal was unclear, its occurrence is nevertheless alarming. These findings suggest that in the absence of more intensive and widespread community sensitisation

the use of oxygen and bCPAP in Malawi may be more effective in units with higher provider-to-parent ratios to facilitate closer monitoring of parental interactions with their children, rather than in general paediatric wards as was done in CPAP IMPACT. District hospitals in Malawi do not have high dependency or intensive care units.

Oxygen support has been shown to be critically important in the care of children with respiratory distress and is included in the WHO list of essential medicines.[19 20] For oxygen to have its full benefit and for implementation of newer technologies such as bCPAP to be effective, there is a clear need for community education. Caregivers displayed a strong support for both oxygen and bCPAP following treatment, even if their child died. This suggests that with exposure and education both can be considered acceptable therapies among caregivers. Caregivers in multiple groups expressed a desire for information in their communities before their children were sick. Community health workers have been shown to be effective in both community management of some diseases and preventative measures.[21] Community health workers delivering preventative care in low- and middle-income countries have been found to be effective in community education.[22] Similar efforts may be effective in increasing education about respiratory support technologies. Additionally, as many mothers commented how they would recommend these therapies to other mothers, mothers who have experienced treatment could be a useful resource for educating their communities.

Throughout focus group discussions, mothers referred to feeding as a surrogate marker for health. They described needing to bring the child to the hospital because they had stopped feeding. The ability of the child to eat was often cited as a benefit of therapy and a way of seeing improvement. There have been strong efforts in Malawi to encourage exclusive breast feeding and their child's ability to eat (breast feed or solids for older children) was clearly important to mothers. This may potentially help explain some of the findings of CPAP IMPACT, which showed increased risk mortality in the bCPAP group, compared with the oxygen group.[15] The authors speculated that these findings could be in part due to a higher risk of aspiration from oral feeding despite recommendations for intravenous fluids or nasogastric feeds while receiving positive pressure ventilation from bCPAP, compared with low flow oxygen. The use of feeding as a marker of health status supports the concern that mothers may have continued to feed orally despite the child's decreased respiratory status and the provider's recommendation against oral feeding.

There are several notable limitations. First, the views of study participants may not be representative of all views. Mothers who participated in a group had already consented to having their child receive therapy and all focus groups were conducted post-intervention. Given we did not collect detailed demographic information from main trial participants, we cannot evaluate whether focus group participants were or were not representative of the

main trial participants. The near universal transition from a negative or fearful opinion to one of acceptance may suggest that only mothers who thought the therapy was beneficial chose to participate in focus group discussions, which would introduce bias into the results. These results should therefore be interpreted within this context. Additionally, groups did not include mothers who never presented to the hospital. This population is harder to reach and would require resources beyond the capacity of this study. However, participants were able to clearly talk about community views and opinions they had heard from their peers and the sample size of women included was sufficient to achieve thematic saturation. Second, the beliefs of participants may not be generalisable to other contexts and oxygen use and bCPAP implementation in other settings will need to consider unique culture viewpoints. Fewer women with deceased children participated in the focus groups. However, thematic saturation was achieved, and this group's views did not differ significantly from caregivers whose children survived. Lastly, in this setting, mothers may be compelled to provide answers that they believe are desired by the research team. We attempted to avoid this by explaining to mothers that all opinions were open and by choosing a facilitator who was independent of clinical care and had never met or interacted with mothers prior. Focus group meetings took place outside the hospital ward in a private room and only Malawians were present to encourage honesty and openness among participants.

## CONCLUSION

In Malawi there are widespread negative beliefs about oxygen and bCPAP therapies with community members often citing that 'oxygen kills babies.' These beliefs may lead to delays in care for children with respiratory disease and may also interrupt care within the hospital. Mothers reported feeling anxious about allowing their child to receive treatment. Widespread community sensitisation is needed to educate communities and improve acceptance of oxygen and other respiratory therapies. Additionally, these results suggest that the role of feeding and its relationship to the effectiveness of respiratory therapies in Malawi needs further evaluation.

**Author affiliations**
[1]Pediatrics, McGaw Medical Center of Northwestern University, Chicago, Illinois, United States
[2]Project Malawi, University of North Carolina System, Lilongwe, Malawi
[3]University of North Carolina Project Malawi, Lilongwe, Malawi
[4]Department of Pediatrics, University of North Carolina at Chapel Hill, Chapel Hill, North Carolina, United States
[5]Infectous Disease, University of North Carolina at Chapel Hill, Chapel Hill, North Carolina, United States
[6]Acute Respiratory Infection Unit, Ministry of Health, Lilongwe, Malawi
[7]Emergency Medicine, Cincinnati Children's Hospital Medical Center, Cincinnati, Ohio, United States
[8]University of Cincinnati College of Medicine, Cincinnati, Ohio, United States
[9]Pediatric Critical Care Medicine, University of Utah, Salt Lake City, Utah, United States
[10]Eudowood Division of Pediatric Respiratory Sciences, Johns Hopkins School of Medicine, Baltimore, Maryland, United States
[11]International Health, Johns Hopkins Bloomberg School of Public Health, Baltimore, Maryland, United States

**Acknowledgements** We thank caregivers and their children for participating in this trial, the Malawi Ministry of Health and the CPAP IMPACT study team for their contributions to this study.

**Contributors** KS, LR, TM and EM participated in study design, data collection and statistical analysis and were major contributors to manuscript writing. MT participated in focus group discussion (FGD) guide preparation, conducted the FGDs, completed translation and transcription and contributed to analysis and manuscript preparation. DK contributed to data collection and design and implementation of CPAP IMPACT. MCH, NL, AGS, ME and EM contributed to study design and implementation of CPAP IMPACT. EM was the principle investigator. All authors contributed to manuscript preparation and approved the final manuscript.

**Funding** KS received funding for this work through a Doris Duke Charitable Foundation grant (2016177) supporting the Doris Duke International Clinical Research Fellows programme at the University of North Carolina Chapel Hill. The Bill & Melinda Gates Foundation (OPP1123419), International AIDS Society (141022) and Health Empowering Humanity provided funding for CPAP IMPACT.

**Competing interests** None declared.

**Patient and public involvement** Patients and/or the public were not involved in the design, or conduct, or reporting or dissemination plans of this research.

**Patient consent for publication** Not required.

**Ethics approval** This study was approved by both the National Health Research Science Committee of Malawi (1325) and the Institutional Review Board of John Hopkins University (IRB00199008).

**Provenance and peer review** Not commissioned; externally peer reviewed.

**Data availability statement** Data are available upon reasonable request. De-identified transcripts of focus group discussions are available on request from corresponding authors.

**ORCID iD**
Kristen L Sessions http://orcid.org/0000-0003-2062-8746

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
