## [Reviewer comments · BMJ Open]

ARTICLE DETAILS

TITLE (PROVISIONAL)	Focus group discussions on low-flow oxygen and bubble CPAP treatments among mothers of young children in Malawi: a CPAP IMPACT sub-study
AUTHORS	Sessions, Kristen; Ruegsegger, Laura; Mvalo, Tisungane; Kondowe, Davie; Tsidyia, Mercy; Hosseinipour, Mina C.; Lufesi, Norman; Eckerle, Michelle; Smith, Andrew; McCollum, Eric

VERSION 1 – REVIEW

REVIEWER	Andrew Argent Department of Paediatrics and Child Health, University of Cape Town and Red Cross War Memorial Children's Hospital South Africa
REVIEW RETURNED	22-Oct-2019

GENERAL COMMENTS	General Comments The authors have addressed an extremely important issue, namely that of the perceptions of care in hospital, by the mothers of infants / children who receive that care. It is important to note that all the participants in this study had provided informed consent to participate in a study comparing the use of low flow oxygen and bubble CPAP. It was possible to adjust the concentration of oxygen delivered on the bCPAP system (although some children would have received bCPAP with oxygen directly from the oxygen concentrator without blended air), and although gas utilized in the bCPAP system would have been humidified, there is limited information available regarding adequacy of humidification. The authors report that participants describe limitation of therapy by intermittent removal of nasal cannulae or masks. It may be useful to have a sense of how well the clinical areas are staffed after hours – there is a suggestion that for at least some of the day, there are no healthcare workers in the clinical areas. Although the study focused on concerns of mothers regarding the use of oxygen therapies, there appears to be an assumption that oxygen therapy is always helpful. While oxygen therapy is clearly life-saving in hypoxaemia, it does seem that there is a risk that unnecessary oxygen therapy may be associated with harm. In at least 1 study the roll out of oxygen therapy in a country was associated with increased mortality in one group of infants. It would be interesting to know if the use of pulse oximetry would be perceived as helpful (or not) to support mothers when seeing if their
---

	children respond to oxygen therapy. Specific Comments Title When I read the title, I assumed that the caregivers were in fact healthcare workers. It may be useful to reconfigure the title to make it clear that the article refers to the “parents” or Mother (as it seems that only mothers were interviewed) and NOT to healthcare workers. An alternative way may be to refer to “caregivers of children admitted to the CPAP IMPACT study in Malawi” Abstract I think that the abstract should include some reference to the fact that all mothers interviewed had previously participated in the CPAP IMPACT study (with informed consent for that study). Introduction The authors set out to identify community knowledge and feelings, but their sample consisted of mothers who had consented to be part of the CPAP IMPACT study. Presumably all those mothers had provided informed consent to participate in the study. It would thus be important to disclose what information was given to the mothers at the time that they provided informed consent to enter the study. Unfortunately the original article reporting on the study methodology (1) does not seem to provide that information in detail, nor does the publication of the data (2). Methods Sampling It is not at all clear to this reader what qualifies someone as an “emancipated minor”. I think that it means someone who is a minor, but is regarded as being an independent adult by virtue of having had an infant.. The authors state that mothers were “contacted via phone”. Is it possible to have a sense of how widespread access to “phone” is in this particular community? Again could this mode of access change the sample from the original cohort? It would be useful to have some sense of how the travel stipend and snack would be regarded as a possible incentive to participate in the study. Patient and public involvement. There is a flat statement that “It was not possible to involve patients or the public in the design ...”. I can understand why infants who received CPAP could not be involved, but I don’t really understand why members of the public could not be involved in the design of the study. Perhaps a short explanation is called for. Results Demographics I am surprised at the age of the mothers in the study, is this reflective of the average age of mothers who have sick infants in the hospital? I had an expectation that the average age might be lower. As another part of that question, were the ages normally distributed, and would median and range not be more appropriate? Perhaps the authors could comment on to what extent the demographics of the study group reflected the demographics of the
--	--

	wider community. Discussion Page 19 Para 1 Throughout this paragraph the authors seem to refer to “feeding” and “breast-feeding” interchangeably. Is it possible that there are different implications between “eating” and “breastfeeding”? It would be interesting to know what options were used in infants who had significant respiratory distress. Were they given intravenous fluids? Were nasogastric tubes placed, and what feeds were utilized? Conclusions It interests me that the authors have not suggested that the mothers who believe that oxygen therapy was helpful for their children might be a useful resource for training in the community. Tables Table 1 The age and number of children are represented as mean (SD), but would be better as median and range if they are not normally distributed (which is very possible). References 1. Smith AG, Eckerle M, Mvalo T, et al. CPAP IMPACT: a protocol for a randomised trial of bubble continuous positive airway pressure versus standard care for high-risk children with severe pneumonia using adaptive design methods. BMJ open respiratory research. 2017;4:e000195 doi: 10.1136/bmjresp-2017-000195 [published Online First: 2017/09/09]. 2. McCollum ED, Mvalo T, Eckerle M, et al. Bubble continuous positive airway pressure for children with high-risk conditions and severe pneumonia in Malawi: an open label, randomised, controlled trial. The Lancet Respiratory Medicine. 2019; doi: 10.1016/s2213-2600(19)30243-7 [published Online].
--	--

REVIEWER	Elisabeth Riviello Beth Israel Deaconess Medical Center and Harvard Medical School, Boston, MA, USA
REVIEW RETURNED	21-Nov-2019

GENERAL COMMENTS	Overall This is a very nicely-designed study with clear outcomes and implications. It is well-written. It has some surprising and important findings, particularly around mothers’ acceptance of therapy even in the setting of a child’s death, as well as the importance of feeding as an indicator of improvement for mothers. A few things that I’d like to see / understand better: - I see really no literature background on other studies like this. I would do a careful literature review for other studies on acceptability of oxygen therapies including ventilation, whether in adults or kids. Even studies that are of this design but about a different medical intervention or in a different area of the world, would be helpful for providing context for the study. It feels as if this study is the only one of its kind, and I suspect there are other qualitative studies with some similarities, which would be good to describe and reference. If they don’t exist, then noting the careful literature review with terms
--

	used, and lack of other studies found would be helpful.  - Anything that can be said about acceptability of low flow or bCPAP in any other setting would be helpful. While this is incredibly interesting, it does leave the reader wondering how widespread these beliefs are, outside of this locale. - The question of bias is still bothersome. In some ways it strengthens the study that the women shared initial negative perceptions of therapy, and these were women who agreed to come in. But the seemingly universal change from negative to positive views on the therapy seems like it could represent the selection who had that change of opinion and were therefore more likely to come in to discuss it. This is a huge limitation and needs to be discussed in more detail. - If oxygen has been available in the community for 18 years, can the authors discuss more of why it might be that perceptions in the community are so negative about oxygen. The authors find that women who experienced a child on oxygen almost universally had a change of opinion to a positive experience in their study. It would seem that if this was the prior experience, this would have trickled into the community over time. Was it that the experience of oxygen and bCPAP during the study itself was more positive than prior experiences with oxygen? Or something else? Again, the bias of who would come in makes me wonder if the change in perceptions noted are a very select group of women. Detailed comments: Title  - I would take “CPAP IMPACT” out of the title since most readers won’t be familiar with that trial Study design  - should be “randomized controlled trial” (not “randomized control trial”) Discussion  - “Oxygen support has been shown to be critically important in the care of children with respiratory distress and is included in the WHO list of essential medicines.” – this could use citations. Trevor Duke has some good studies showing the benefit of oxygen in pediatric pneumonia, for example.
--	---

VERSION 1 – AUTHOR RESPONSE

Reviewer: 1

Reviewer Name: Andrew Argent

Institution and Country:

Department of Paediatrics and Child Health,

University of Cape Town and Red Cross War Memorial Children's Hospital

South Africa

Please state any competing interests or state ‘None declared’: None declared

Please leave your comments for the authors below

General Comments

The authors have addressed an extremely important issue, namely that of the perceptions of care in

hospital, by the mothers of infants / children who receive that care.

REVIEWER:

It is important to note that all the participants in this study had provided informed consent to participate in a study comparing the use of low flow oxygen and bubble CPAP. It was possible to adjust the concentration of oxygen delivered on the bCPAP system (although some children would have received bCPAP with oxygen directly from the oxygen concentrator without blended air), and although gas utilized in the bCPAP system would have been humidified, there is limited information available regarding adequacy of humidification.

AUTHORS:

Please note that although beyond the scope of this current write-up, which is specifically reporting the results of focus group discussions with mothers, the bCPAP system we used did not allow adjustment of the concentration of oxygen as we were using an Airsep NewLife Intensity oxygen concentrator, which did not have this functionality.

REVIEWER:

The authors report that participants describe limitation of therapy by intermittent removal of nasal cannulae or masks. It may be useful to have a sense of how well the clinical areas are staffed after hours – there is a suggestion that for at least some of the day, there are no healthcare workers in the clinical areas.

AUTHORS:

We appreciate this comment. While beyond the scope of this manuscript – which is specifically reporting the results of focus group discussions with mothers and not the primary trial results – we have clarified staffing in the discussion section (line 340-341). For the reviewer's benefit there were never periods of time where there were no health care workers present but the staffing levels were lower overnight, as is typical in all healthcare settings, which means staff may be in a different part of the ward or caring for other children.

REVIEWER:

Although the study focused on concerns of mothers regarding the use of oxygen therapies, there appears to be an assumption that oxygen therapy is always helpful. While oxygen therapy is clearly life-saving in hypoxaemia, it does seem that there is a risk that unnecessary oxygen therapy may be associated with harm. In at least 1 study the roll out of oxygen therapy in a country was associated with increased mortality in one group of infants.

AUTHORS:

We appreciate this comment and agree that no therapy, oxygen included, is risk free. Notably, at the time of this study no such data existed (that we were aware of) regarding the possible risks of oxygen treatment among children in low-income and middle-income countries. In the main trial report we have discussed this issue in detail. Nonetheless, we have added a sentence to the methods (line 121) clarifying that all children enrolled met criteria for oxygen therapy based on WHO guidelines, which is the current standard of care in Malawi given pulse oximeters are not routinely available in all facilities.

REVIEWER:

It would be interesting to know if the use of pulse oximetry would be perceived as helpful (or not) to support mothers when seeing if their children respond to oxygen therapy.

AUTHORS:

We agree that this would be interesting to investigate further. However, in this trial pulse oximetry was not systemically used to provide feedback to participating mothers and so we cannot comment

further.

REVIEWER:

Specific Comments

Title

When I read the title, I assumed that the caregivers were in fact healthcare workers. It may be useful to reconfigure the title to make it clear that the article refers to the “parents” or Mother (as it seems that only mothers were interviewed) and NOT to healthcare workers. An alternative way may be to refer to “caregivers of children admitted to the CPAP IMPACT study in Malawi”

AUTHORS:

Thank you, we have used the term ‘mothers’ in the new title to clarify this point.

REVIEWER:

Abstract

I think that the abstract should include some reference to the fact that all mothers interviewed had previously participated in the CPAP IMPACT study (with informed consent for that study).

AUTHORS:

Thank you for this comment, we have added the phrase “with informed consent” and the title of the trial to the abstract (lines11-12).

REVIEWER:

Introduction

The authors set out to identify community knowledge and feelings, but their sample consisted of mothers who had consented to be part of the CPAP IMPACT study. Presumably all those mothers had provided informed consent to participate in the study. It would thus be important to disclose what information was given to the mothers at the time that they provided informed consent to enter the study. Unfortunately the original article reporting on the study methodology (1) does not seem to provide that information in detail, nor does the publication of the data (2).

AUTHORS:

Thank you for this comment, we have also listed information reviewed during the main trial informed consent process. We have also added information to the study design section describing the different sources of information mothers were asked about during the FGDs including community beliefs, information taught in the consent process, and experiential knowledge gained during their child’s treatment (lines 127-130).

REVIEWER;

Methods

Sampling

It is not at all clear to this reader what qualifies someone as an “emancipated minor”. I think that it means someone who is a minor, but is regarded as being an independent adult by virtue of having had an infant.

AUTHORS:

We have replaced the phrase “emancipated minors” with “women who were under 16 years old but had a child were allowed to participate in accordance with ethical research standards and laws in Malawi” to clarify.

REVIEWER: The authors state that mothers were “contacted via phone”. Is it possible to have a sense of how widespread access to “phone” is in this particular community? Again could this mode of

access change the sample from the original cohort?

AUTHORS:

About 50% of Malawian population may be reached by phone from experience in the CPAP IMPACT trial (when performing the day 30 telephonic visit). Generally, those who live further from the townships are unlikely to have phones or phone use. Although it should also be noted that due to electric power outages, some families may also not be reachable on attempt of telephonic contacts since the phone's batteries may have died.

REVIEWER:

It would be useful to have some sense of how the travel stipend and snack would be regarded as a possible incentive to participate in the study.

AUTHORS:

It is challenging to quantify how much of an incentive this was for women but we based the amount of travel stipend and snack given on standard practices across many studies in Malawi and per Malawi Institutional review board guidelines, which accounts for average transportation costs by bike taxi or public bus. We have clarified this in line 143.

REVIEWER:

Patient and public involvement.

There is a flat statement that "It was not possible to involve patients or the public in the design ...". I can understand why infants who received CPAP could not be involved, but I don't really understand why members of the public could not be involved in the design of the study. Perhaps a short explanation is called for.

AUTHORS:

This was primarily due to resource limitations for this study (financial and time). We did not have the resources of larger trials to do widespread community engagement activities before the study. However, Salima District Hospital hospital administration and the Malawi Ministry of Health were closely involved in both this sub-study and the larger CPAP IMPACT trial design and execution. (lines 163-165) It is also of note that one of the primary goals of this specific research was to get feedback of participants, who are representatives of the community, so that future research and care can be informed.

REVIEWER:

Results

Demographics

I am surprised at the age of the mothers in the study, is this reflective of the average age of mothers who have sick infants in the hospital? I had an expectation that the average age might be lower. As another part of that question, were the ages normally distributed, and would median and range not be more appropriate?

AUTHORS:

Thank you for this comment. Unfortunately, we do not have data about the age of mothers in the wider study or the hospital as a whole so are not able to comment on whether this is representative or not. The ages of the mothers was not normally distributed, so we have corrected this to median and interquartile range. Thank you for suggesting this.

REVIEWER:

Perhaps the authors could comment on to what extent the demographics of the study group reflected the demographics of the wider community.

AUTHORS:

Thank you for this comment. Unfortunately, we do not have data about the age of mothers in the wider study or the hospital as a whole so are not able to comment on whether this is representative or not. The hospital only collects data on patients, not caregiver demographics. We have added this as a limitation in the discussion section.

REVIEWER:

Discussion

Page 19 Para 1

Throughout this paragraph the authors seem to refer to “feeding” and “breast-feeding” interchangeably. Is it possible that there are different implications between “eating” and “breastfeeding”?

AUTHORS:

We have tried to make sure that only the term feeding is used throughout the paragraph, except for in the sentence showing that breastfeeding was emphasized. We feel that this distinction is important because it could highlight some mixed messaging mothers are receiving around breastfeeding and feeding. We have clarified that feeding includes both breast feeding and solid foods for older children as children up to 59 months of age were included in the study.

REVIEWER:

It would be interesting to know what options were used in infants who had significant respiratory distress. Were they given intravenous fluids? Were nasogastric tubes placed, and what feeds were utilized?

AUTHORS:

Both IV fluids and NG tubes were utilized in the study. However, comments by mothers suggest that oral feeding persisted despite these interventions. This has been clarified in line 377.

REVIEWER:

Conclusions

It interests me that the authors have not suggested that the mothers who believe that oxygen therapy was helpful for their children might be a useful resource for training in the community.

AUTHORS:

Thank you for this excellent point. This is something the mothers mentioned themselves and we discussed while analyzing the data. We have added a sentence to the discussion to make this explicit (line 367-369).

REVIEWER:

Tables

Table 1

The age and number of children are represented as mean (SD), but would be better as median and range if they are not normally distributed (which is very possible).

AUTHORS: As previously indicated the age of mothers is not normally distributed and we have corrected this to the summary statistic of median and interquartile range. The number of children per mother is normally distributed and so we have retained mean and standard deviation. We had not previously reported the age of participant children but we have added this into Table 1 now. The distribution of the age of participant children was not normally distributed and so we have used the summary statistic of mean and interquartile range to describe this variable. Statistical analyses for

quantitative variables used mean and standard deviation to describe normally distributed data and median and interquartile range to describe not normally distributed data. We compared means of normally distributed data using t tests and medians of not normally distributed data using the Wilcoxon-Mann-Whitney test. Quantitative analyses were conducted using Stata 15.1 (College Station, USA).

References

1. Smith AG, Eckerle M, Mvalo T, et al. CPAP IMPACT: a protocol for a randomised trial of bubble continuous positive airway pressure versus standard care for high-risk children with severe pneumonia using adaptive design methods. *BMJ open respiratory research*. 2017;4:e000195 doi: 10.1136/bmjresp-2017-000195 [published Online First: 2017/09/09].
2. McCollum ED, Mvalo T, Eckerle M, et al. Bubble continuous positive airway pressure for children with high-risk conditions and severe pneumonia in Malawi: an open label, randomised, controlled trial. *The Lancet Respiratory Medicine*. 2019; doi: 10.1016/s2213-2600(19)30243-7 [published Online.

Reviewer: 2

Reviewer Name: Elisabeth Riviello

Institution and Country: Beth Israel Deaconess Medical Center and Harvard Medical School, Boston, MA, USA

Please state any competing interests or state 'None declared': None declared

Please leave your comments for the authors below

Overall

This is a very nicely-designed study with clear outcomes and implications. It is well-written. It has some surprising and important findings, particularly around mothers' acceptance of therapy even in the setting of a child's death, as well as the importance of feeding as an indicator of improvement for mothers.

REVIEWER:

A few things that I'd like to see / understand better:

- I see really no literature background on other studies like this. I would do a careful literature review for other studies on acceptability of oxygen therapies including ventilation, whether in adults or kids. Even studies that are of this design but about a different medical intervention or in a different area of the world, would be helpful for providing context for the study. It feels as if this study is the only one of its kind, and I suspect there are other qualitative studies with some similarities, which would be good to describe and reference. If they don't exist, then noting the careful literature review with terms used, and lack of other studies found would be helpful. Anything that can be said about acceptability of low flow or bCPAP in any other setting would be helpful. While this is incredibly interesting, it does leave the reader wondering how widespread these beliefs are, outside of this locale.

AUTHORS:

A literature search was completed as background for the study design and manuscript write up. We cite four relevant studies that provided background information. We tried to focus our references on pediatric populations and the experiences of mothers and caregivers and, when possible, in an African context. A full scoping review or systematic review of the literature extends beyond the scope of this manuscript.

11. Gondwe MJ, Gombachika B, Majamanda MD. Experiences of caregivers of infants who have been on bubble continuous positive airway pressure at Queen Elizabeth Central Hospital, Malawi: A descriptive qualitative study. *Malawi Medical Journal* 2017;29(1):5-10.
12. Peeler A, Fulbrook P, Kildea S. The experiences of parents and nurses of hospitalised infants requiring oxygen therapy for severe bronchiolitis: A phenomenological study. *Journal of Child Health Care* 2015;19(2):216-28.
13. Foster J, Bidewell J, Buckmaster A, et al. Parental stress and satisfaction in the non-tertiary special care nursery. *Journal of advanced nursing* 2008;61(5):522-30.
14. Cervantes AC, Feeley N, Lariviere J. The experience of mothers whose very low-birth-weight infant requires the delivery of supplemental oxygen in the neonatal intensive care unit. *Advances in Neonatal Care* 2011;11(1):54-61.

REVIEWER:

- The question of bias is still bothersome. In some ways it strengthens the study that the women shared initial negative perceptions of therapy, and these were women who agreed to come in. But the seemingly universal change from negative to positive views on the therapy seems like it could represent the selection who had that change of opinion and were therefore more likely to come in to discuss it. This is a huge limitation and needs to be discussed in more detail.

AUTHORS:

We agree this is a significant limitation of this study and focus group studies in general. We invited all women regardless of outcome or intervention but the population who chose to participate could certainly be skewed toward women who perceived the interventions as positive. We have added a sentence to clarify that we recognize this is a limitation and selective participation may have altered the results of the study. (lines 387-389)

REVIEWER:

- If oxygen has been available in the community for 18 years, can the authors discuss more of why it might be that perceptions in the community are so negative about oxygen. The authors find that women who experienced a child on oxygen almost universally had a change of opinion to a positive experience in their study. It would seem that if this was the prior experience, this would have trickled into the community over time. Was it that the experience of oxygen and bCPAP during the study itself was more positive than prior experiences with oxygen? Or something else? Again, the bias of who would come in makes me wonder if the change in perceptions noted are a very select group of women.

AUTHORS:

This is an excellent point and we did wonder why such persistent negative opinions persisted. It was noted, however, that very few of our participants had first hand exposure (oxygen therapy use themselves or a family member). Those who had encountered oxygen before had significantly more positive opinions at the start of treatment compared to those who had no previous exposure. This may strengthen the idea that exposure can change perceptions but there were not enough women who had encountered these therapies before to draw broader conclusions. We have tried to clarify the viewpoint of these few women in several sections of the manuscript (lines 228-229, 329-331).

REVIEWER:

Detailed comments:

Title

- I would take "CPAP IMPACT" out of the title since most readers won't be familiar with that trial

AUTHORS:

We have moved this to the end to clarify that it was a sub-study. We feel it is important to highlight that this was in the context of a larger study because this did shape the type of exposure the mothers had and which mothers were included.

REVIEWER:

Study design

- should be “randomized controlled trial” (not “randomized control trial”)

AUTHORS:

We have corrected this typo.

REVIEWER:

Discussion

- “Oxygen support has been shown to be critically important in the care of children with respiratory distress and is included in the WHO list of essential medicines.” – this could use citations. Trevor Duke has some good studies showing the benefit of oxygen in pediatric pneumonia, for example.

AUTHORS:

We have added two citations for this:

19. World Health Organization Model List of Essential Medicines, 21st List, 2019. Geneva: World Health Organization; 2019. Licence: CC BY-NC-SA 3.0 IGO.

20. Rojas-Reyes MX, Granados Rugeles C, Charry-Anzola LP. Oxygen therapy for lower respiratory tract infections in children between 3 months and 15 years of age. Cochrane Database of Systematic Reviews 2014, Issue 12. Art. No.: CD005975. DOI: 10.1002/14651858.CD005975.pub3.

VERSION 2 – REVIEW

REVIEWER	Andrew Argent University of Cape Town and Red Cross War Memorial Children's Hospital
REVIEW RETURNED	28-Mar-2020
GENERAL COMMENTS	I am grateful for the changes made in response to the reviewers. I do not want any further revisions.